# Rectal Biopsy for Hirschsprung’s Disease: A Multicentre Study Involving Biopsy Technique, Pathology and Complications

**DOI:** 10.3390/children10091488

**Published:** 2023-08-31

**Authors:** Gil Vervloet, Antoine De Backer, Stijn Heyman, Paul Leyman, Sebastiaan Van Cauwenberge, Kim Vanderlinden, Charlotte Vercauteren, Dirk Vervloessem, Marc Miserez

**Affiliations:** 1Universitair Ziekenhuis Leuven, Katholieke Universiteit Leuven, 3000 Leuven, Belgium; 2Universitair Ziekenhuis Brussel, KidZ Health Castle, Saffier Network, 1000 Brussels, Belgium; 3Ziekenhuis Netwerk Antwerpen, Ziekenhuis aan de Stroom, Queen Paola Children’s Hospital, Saffier Network, 2650 Edegem, Belgium; 4Gasthuiszusters Antwerpen, Ziekenhuis aan de Stroom, Saffier Network, 2000 Antwerpen, Belgium; 5Algemeen Ziekenhuis Sint-Jan, 8000 Brugge, Belgium

**Keywords:** Hirschsprung’s disease, rectal biopsy, suction biopsy, punch biopsy, pathology Hirschsprung

## Abstract

Background: The heterogeneity of rectal biopsy techniques has encouraged us to search for a surgical and pathological standardisation of this diagnostic technique to exclude Hirschsprung’s disease. The different amounts of information on the anatomopathology report prompted us to compile a template for the anatomopathology report for diagnostic rectal biopsies for surgical colleagues and pathologists working on Hirschsprung’s disease. Methods: We gathered the anonymous biopsy information and its pathology information from five hospitals for all patients in which rectal biopsies were taken to diagnose Hirschsprung’s disease over two years (2020–2021). Results: Of the 82 biopsies, 20 suction (24.4%), 31 punch (37.8%) and 31 open biopsies (37.8%) were taken. Of all biopsies, 69 were conclusive (84.2%), 13 were not (15.8%). In the suction biopsy group, 60% were conclusive and 40% were not; for punch biopsy, the values were 87% and 13%, respectively and for open biopsy, 97% and 3%. Inconclusive results were due to insufficient submucosa in 6/8 suction biopsies, 4/4 punch biopsies and 0/1 open biopsies. An insufficient amount of submucosa was the reason for an inconclusive result in 6/20 cases (30%) after suction biopsy, 4/31 (12.9%) cases after punch biopsy and 0 cases (0%) after open biopsy. We had one case with major postoperative bleeding post suction biopsy; there were no further adverse effects after biopsy. Conclusions: Diagnostic rectal biopsies in children are safe. Non-surgical biopsies are more likely to give inconclusive results due to smaller amounts of submucosa present in the specimen. Open biopsies are especially useful when previous non-surgical biopsies are inconclusive. An experienced pathologist is a key factor for the result. The anatomopathology report should specify the different layers present in the specimen, the presence of ganglion cells and hypertrophic nerve fibres, their description and a conclusion.

## 1. Introduction

Hirschsprung’s disease (HD) is a congenital condition of the enteric nervous system first described in 1886 by Harald Hirschsprung. The disordered caudal migration of neural crest cells results in a lack of intrinsic innervation (neuronal ganglion cells and enteric glial cells) in the affected intestine. Because of this lack of innervation, there is unrelieved contraction of the colonic wall and inefficient or absent peristalsis that manifests as a functional intestinal obstruction [1].

Usually, we see clinical manifestations of the disease in the first month (65%) or in the first year (95%) after birth. The most frequent manifestations are delayed passage of meconium, abdominal distention, bilious vomiting and severe constipation. In 10–25% of neonates, HD initially presents with serious complications such as Hirschsprung-associated enterocolitis.

The disease is quite rare with an incidence of 1 in 5000 live births and a 4/1 male/female ratio [2]. There are different forms of HD, depending on the extent of the aganglionosis: the classical short-segment form (80–85%) with aganglionosis from the anal canal up to the sigmoid, the long-segment form with aganglionosis up to the caecum, total colonic form and small or total intestinal HD with aganglionosis extending beyond the terminal ileum.

Diagnosis of the disease can be difficult, particularly in premature children. Radiological diagnosis can be done by contrast enema. It allows us to rule out other differential diagnosis of lower gastro-intestinal obstructions and try to assess the length of the affected aganglionic segment. The presence of a transition zone and the rectosigmoidal index being less than one, are sensitive and specific signs of Hirschsprung disease. However age (neonates) and extent (long segment) are limiting factors in the diagnostic value of contrast enema [3,4,5,6,7,8].

Anorectal manometry is another tool used to diagnose HD. It can demonstrate an absent rectosphincteric reflex in HD patients but with false negative and false positive results due to technical problems, low internal anal pressure, megarectum or body movements [4,9]. 

Histology is still the gold standard to diagnose HD. There are a variety of different rectal biopsy techniques: suction biopsy, punch biopsy and open biopsy. Clinical practice varies, but most practitioners obtain two to three rectal biopsy specimens per procedure to exclude inconclusive results [10]. The specimen is sent to the pathology department immediately after biopsy. There, the specimen is fresh frozen or fixed, depending on the additional histochemistry used, and stained with haematoxylin–eosin (HE) to demonstrate aganglionosis and nerve trunk hypertrophy. Ideally, one-third of the specimen is submucosa. The presence of one ganglion cell excludes HD.

The heterogeneity of rectal biopsy techniques, both in surgical harvesting and pathological examination and reporting, used in our hospitals, encouraged us to compare techniques, evaluate complications and search for a standard for reporting the histological results in an easy-to-use format to facilitate decision-making concerning further (surgical) diagnostics and therapy.

This study is designed to assess the hypothesis that the technique used for a rectal biopsy can affect the percentage of inconclusive results. We expect to see no difference in inconclusive results after biopsy between the punch biopsy versus suction biopsy, but still see better results in the open biopsy group. We expect this can be attributed to the amount of tissue (mucosa and submucosa) in the biopsy. If possible, we will standardize our diagnostic approaches concerning biopsies. Furthermore, we would like to compile a checklist concerning the anatomopathology report for diagnostic rectal biopsies for surgical colleagues and pathologists working on HD.

## 2. Materials and Methods

### 2.1. Patients

We gathered the anonymous biopsy information from five hospitals (UZ Leuven, UZ Brussel, GZA Antwerpen, ZNA Antwerpen and AZ Sint-Jan Brugge/Oostende) for all patients in which rectal biopsies were taken to exclude or confirm Hirschsprung’ disease over two years (2020–2021). This was performed in a retrospective manner.

### 2.2. Data

Information that was shared involved the type of biopsy (suction, punch or open), the age of the patient at the time of biopsy, information concerning pathology (ganglion cells, nerve hypertrophy, layer in which they were found) and the result (conclusive or not). If the biopsies were inconclusive, we registered the cause of why they were inconclusive and the type of biopsy used during the second attempt. Furthermore, all adverse effects of the procedure were registered.

In cases where information was incomplete, we did not include the patient in this study.

### 2.3. Statistical Analysis

Fisher’s exact test was used to compare proportions between two groups. The *p*-value from the extension of Fisher’s exact test (the Fisher–Freeman–Halton test) was reported for the comparison of three groups. *p*-values smaller than 0.05 were considered as significant. Since no corrections for multiple testing were applied, a single significant *p*-value needs to be interpreted with caution. All analyses were performed using SAS software, version 9.4 of the SAS System for Windows.

### 2.4. Biopsy Techniques

#### 2.4.1. Suction Biopsy (“Non-Surgical”)

There are many techniques and devices to obtain a suction biopsy. In general (mostly depending on the age of the child being less than six months old), the patient is awake in the lithotomy position. The biopsy is taken along the posterior wall of the rectum around two centimetres above the dentate line. No anterior biopsies are taken because of the possibility of perforation into the vaginal wall or abdominal cavity. Mostly, two or three biopsies are taken each time. In the results section, we describe the result of one intervention in one patient, even though multiple biopsies could have been taken during the intervention.

Most biopsies are taken with the SOLO-RBT^®^ instrument (SAMO Biomedica, Bologna, Italy), a modified version of the original device proposed by Noblett [11]. This instrument is designed to be operated by one person. A pressure range from 0 to 760 cm H_2_O is used to suck in the mucosa and a part of the submucosa. One push on the trigger cuts the sucked-in tissue. Depending on the amount of tissue sampled, the pressure is adjusted for the next biopsy [12].

Some of the biopsies are taken with the rbi2 system^®^ (Aus Systems, Allenby Gardens, Australia). The pressure can be seen on a manometer and is ideally 150 cm H_2_O. The device uses a lower pressure than the older systems (SOLO-RBT^®^), but it must be handled by two people. One person positions the instrument in the rectum, the other applies the pressure by withdrawing the plunger to 3 to 5 mL. When the trigger is pulled, the tissue that is sucked in is cut and collected in the tip of the instrument. Depending on the amount of tissue sampled, the pressure is adjusted for the next biopsies. Caution should be taken not to exceed the recommended suction pressure.

#### 2.4.2. Punch Biopsy (“Non-Surgical”)

The punch biopsies are performed mostly without general anaesthesia, although this depends on age (under one year of age), in the lithotomy position. The biopsy is also taken along the posterior wall of the rectum. Depending on the age of the patient, various sizes of biopsy forceps are used. With the forceps, a small piece of mucosa and submucosa is taken and pulled out. Mostly, two or three biopsies are taken each time. In the results section, we describe the result of one intervention in one patient, even though multiple biopsies could have been taken during the intervention.

#### 2.4.3. Open Biopsy (“Surgical”)

Open rectal biopsies are performed under general anaesthesia in the lithotomy position. An anuscope or speculum is used to allow good vision of the posterior wall of the rectum. A deep biopsy with mucosa and submucosa of approximately one centimetre wide and two centimetres long is taken, starting one and a half centimetres above the dentate line. The mucosa is afterwards closed with separate stitches of vicryl 4/0. Open biopsies are mostly used when the less-invasive techniques described above did not give conclusive results or when the child is too old to perform a biopsy without general anaesthesia.

### 2.5. Adverse Effects

Most adverse effects (bleeding, rectal perforation) after biopsy occur immediately after the procedure. When no major bleeding, fever or pain was present, patients were discharged after several hours of observation.

### 2.6. Anatomopathology

Nerve hypertrophy and the absence of ganglion cells are pathological signs for the diagnosis of HD. We can see nerve hypertrophy and ganglion cells in the Meissner and Auerbach plexus, but one of the plexuses is sufficient for the pathologist to diagnose HD. The Meissner plexus is located in the submucosa and the Auerbach plexus in the muscularis propria, between the circular and longitudinal muscle. Sometimes, offshoots of the nerve hypertrophy from the Meissner plexus can be seen extending in the mucosa layer (lamina propria) on additional histochemistry, and this increases with age [13]. Depending on the pathologist, this can be enough to conclude for HD in superficial biopsies with insufficient submucosa.

Immediately after biopsy the specimen is sent to the pathology department, the specimen is fresh frozen or fixed, depending on the additional histochemistry used, and stained with HE. Additional histochemistry is used to further investigate the specimen. Depending on the hospital, this may be immunohistochemistry or enzyme histochemistry. Immunohistochemistry with calretinin, S100 and synaptophysin has to be carried out on formalin-fixed paraffin-embedded tissue and is used in four out of five hospitals. Additional enzyme histochemistry staining with acetylcholinesterase (AChE) and NADH diaphorase has to be performed on fresh frozen specimens and is used in one hospital. Calretinin (positive in the control sample), S100 (positive in the control sample), synaptophysin (positive in the control sample) and AChE staining (negative in the control sample) highlight the nerve hypertrophy while NADH diaphorase and calretinin place the emphasis on the ganglion cells in the specimen [14,15,16,17,18,19,20].

The pathologist examines the sample and classifies it as conclusive or inconclusive. Conclusive means that the sample is pathological (HD) or normal. A classification of “inconclusive” means the sample is inadequate to make a clear statement about the possible underlying disease.

## 3. Results

In two cases, the information was incomplete. We did not include the patients in this study.

All five hospitals use the open surgical biopsy technique. For the non-surgical biopsies, three out of five use the punch biopsy technique and three use the suction technique, with one hospital using both punch and suction biopsy techniques, depending on the choice of the surgeon. In total, 82 biopsies were taken, of which there were 20 suction (24.4%), 31 punch (37.8%) and 31 open biopsies (37.8%). The mean age at time of biopsy was 194 days in the suction biopsy group, 830 days for punch biopsy and 682 days for open biopsy.

Of all biopsies, 70 were first biopsies (Figure 1). Of those first biopsies, 48 (68.5%) were non-surgical biopsies and 22 (31.5%) were surgical biopsies. In the non-surgical biopsy group, we report 11 inconclusive results (23%) and 37 conclusive results (77%) (16 with the diagnosis of HD (33.3%) and 21 (43.7%) normal biopsies). In the surgical biopsy group, we report 1 (4.5%) inconclusive result and 21 (95.5%) conclusive results (8 (36.4%) with the diagnosis of HD and 13 (59.1%) normal biopsies). The non-surgical biopsy group can be divided in 18 suction biopsies and 30 punch biopsies. Of those 18 suction biopsies, 7 (38.9%) biopsies gave inconclusive results and 11 (61.1%) conclusive results (5 (27.8%) biopsies with the diagnosis of HD and 6 (33.3%) normal biopsies). In the punch biopsy group, we report 4 (13.3%) inconclusive results and 26 (86.6%) conclusive results (11 (36.7%) with the diagnosis of HD and 15 (50%) normal biopsies) (Figure 2).

In total, we have 12 inconclusive first biopsies (17.1%) for which a second biopsy was needed. Of those 12 biopsies, 9 were surgical and 3 non-surgical biopsies. In the non-surgical biopsy group, we had 1 inconclusive result (a suction biopsy), 1 biopsy with the diagnosis of HD (punch) and 1 normal biopsy (suction). In the surgical group, we have 0 inconclusive results, 5 with the diagnosis of HD and 4 normal biopsies (Figure 3).

Of all biopsies, 69 were conclusive (84.2%), 13 were not (15.8%). In the suction biopsy group, 60% were conclusive and 40% were not, for punch biopsy the values were 87% and 13%, respectively, and for open biopsy, the values were 97% and 3%, respectively (Figure 4). Inconclusive results were due to insufficient submucosa in 6/8 suction biopsies, 4/4 punch biopsies and 0/1 open biopsies (Figure 5). For all biopsies, suction biopsies result in an insufficient amount of submucosa in 6/20 cases (30%) and punch biopsies in 8/31 cases (26%). In 4/8 cases of insufficient submucosa after punch biopsy, there was an insufficient amount of submucosa, but it was still enough to achieve a conclusive result. An insufficient amount of submucosa was the reason for an inconclusive result in 6/20 cases (30%) after suction biopsy and 4/31 (12.9%) after punch biopsy (Figure 6).

The difference in conclusive results between the open biopsy group and the non-surgical biopsy group is significant (*p* = 0.014). Furthermore, the difference between these two groups regarding the presence of a sufficient amount of submucosa is also significant (*p* ≤ 0.001) (Table 1). When comparing the three groups we see significant differences in conclusive results between the open biopsies vs. suction biopsies and punch vs. suction biopsies. The presence of a sufficient amount of submucosa was significantly different between the open biopsies vs. punch biopsies and the open biopsies vs. suction biopsies (Table 2).

We had one case with major postoperative bleeding post suction biopsy; there were no further adverse effects after biopsy.

## 4. Discussion

Hirschsprung’s disease (HD) is a rare disease and the gold standard for its diagnosis is pathological confirmation by rectal biopsy. The heterogeneity of rectal biopsy techniques used not only in the world but also in our five centres in which we conducted our study, encouraged us to compare our techniques, evaluate complications and search for a standard for reporting the histological results in an easy-to-use format to facilitate decision-making concerning further (surgical) diagnostics and therapy.

### 4.1. Biopsy

In our study, non-surgical biopsies are prone to yield more inconclusive results. This difference can be explained by the smaller amount of submucosa in the non-surgical biopsy group. This is due to the biopsy technique used and the surgeon taking the (non-surgical) biopsy and the pathologist evaluating it. Both suction (30%) and punch (26%) had around the same proportion of insufficient submucosa in biopsies, but a different number of inconclusive results. It is striking that four out of eight punch biopsies with insufficient submucosa are still evaluated as conclusive and none for suction biopsies. These results are attributed only to the pathologist evaluating the biopsies and not to the biopsy technique used or the surgeon taking the biopsy, with those four conclusive punch biopsies with insufficient submucosa being performed in two hospitals. Sometimes offshoots of the nerve hypertrophy from the Meissner plexus can be seen extending into the mucosa layer (lamina propria) by additional histochemistry. Depending on the expertise of the pathologist, this can be enough to conclude HD. Unfortunately, the different types of biopsies were not evenly distributed by the hospital and consequently not by pathologist. As stated earlier, the additional staining on the specimens is conducted depending on the hospital and the experience of their staff and are therefore not similar. These are clear limitations of our study. A systematic review and meta-analysis performed by Comes et al., reviewing all biopsy techniques for the diagnosis of HD, found no difference in conclusive results between suction (88%), punch (95%) and open techniques (94%) [21]. However, they did not specify if there was sufficient amount of submucosa present in the specimens. A systematic review of the literature concerning suction biopsies for the diagnosis of HD was performed by Friedmacher et al. [22]. They reported 89.9% of adequate tissue samples in suction biopsies for the diagnosis of HD from 14,053 samples. In addition, in this systematic review, there was no documentation about the amount of submucosa present in the specimens. The conclusive results for suction biopsy in these two systematic reviews are clearly higher than in our small series, where only 60% of samples were conclusive (in contrast to 87% for punch biopsies). The first is probably related to surgical technique and/or pathological expertise, but the exact reason for this in our cohort is not clear at this time.

Non-surgical biopsies are biopsies that can be performed without general anaesthesia and are therefore the first choice in neonates. Open biopsy needs general anaesthesia, which is not favourable in neonates. The higher mean age in the suction biopsy group, and certainly in the punch biopsy group, is due to a couple of outliers with higher ages raising the mean age in this small sample group. The number of patients in each biopsy group in our study is too small to stratify the results by age, so we cannot make conclusions on the relationship between age or prematurity and inconclusive biopsies. However, Green et al. showed a substantially and significantly higher percentage of inadequate procedures in the suction biopsy group (24.1%) in comparison with the open biopsy technique (0.9%) in patients older than 6 months. The proportion of inadequate biopsies for those <6 months was 7.6% compared to 25% in those >6 months. Potential explanations for the increase in inadequacy observed in suction biopsies in patients over 6 months include patient cooperation (as suction biopsies are performed without sedation, and larger infants are less likely to comply with the procedure). They explain that their findings may also be due to the decreased density of ganglion cells in the submucosa, occurring with age and a larger rectal vault, as seen in older children, especially those with chronic constipation, which makes it more difficult to perform a suction biopsy. This may necessitate larger biopsies in older children to achieve an equivalent adequate result [23,24].

In our study, we had one complication of a postoperative bleed in the suction group. The number of patients per biopsy technique is too small to make a conclusion about the complication rate difference between biopsy groups. However, from our study and also in the literature, it is clear that all biopsy techniques are safe to perform [21,22,25]. Comes et al. demonstrated a significantly higher complication rate for punch biopsies than for suction biopsies. They attribute the difference between suction and punch biopsies to the fact that the punch biopsy results in larger fragments [21]. However, the complication rates in the open biopsy group were not significantly different compared to the suction or punch biopsy groups.

### 4.2. Anatomopathology Report

Our group gathered information from different hospitals during this study. The amount of information concerning the pathology result was not equivalent across different hospitals, and also differed depending on the pathologist writing the report. There are a few key findings that we, as surgeons, would like to know about the biopsy, with respect to pathology, to assess the next step in the patient’s treatment. Therefore, we compiled a checklist concerning the anatomopathology report for diagnostic rectal biopsies for surgical colleagues and pathologists working on HD.

The pathology report should contain:The different layers of the rectal wall present in the biopsy;The presence of ganglion cells, the layer they are found in and the description of their pattern: regular or irregular;The presence of hypertrophic nerve fibres, the layer or plexus they are found in, their size and the description of further growth in the lamina propria or not;A conclusion from the pathologist: if the result is not conclusive, a short explanation on why this is not the case.

How to stain the specimen is debatable and not part of this study. Immunohistochemistry and enzyme histochemistry are feasible and can both be used for further examination of the specimen. The pathologist should use the technique that he/she is the most familiar with. In general, immunohistochemistry is used on a larger scale because it is easier to work with and can be performed on formalin-fixed paraffin-embedded tissue instead of fresh frozen tissue.

## 5. Conclusions

Diagnostic rectal biopsies in children are safe. The obtained result (conclusive vs. inconclusive) is related to patient factors, but largely to the biopsy technique used, the surgeon performing the biopsy and the pathologist evaluating the biopsy. Non-surgical biopsies are less likely to give conclusive results in comparison with surgical biopsies. Open biopsies are very useful when previous non-surgical biopsies are inconclusive. An experienced pathologist is a key factor in the result. The anatomopathology report should specify the different layers present in the specimen, the presence of ganglion cells and hypertrophic nerve fibres, their description and a conclusion.

## Figures and Tables

**Figure 1 children-10-01488-f001:**
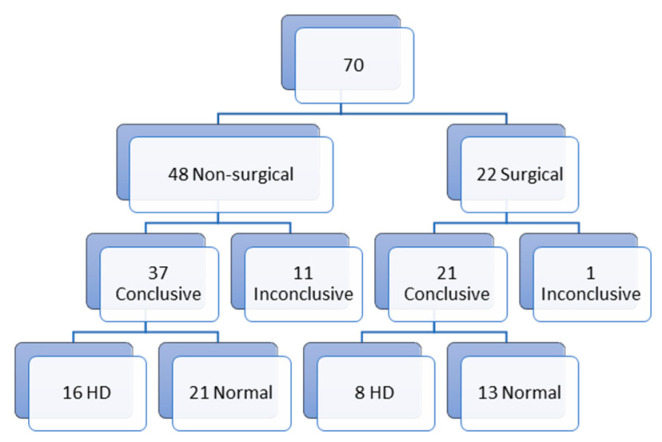
First biopsies.

**Figure 2 children-10-01488-f002:**
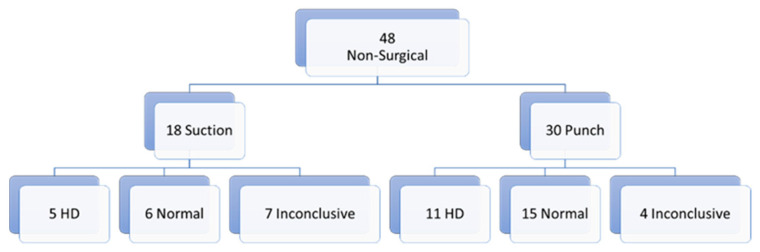
First biopsies: non-surgical.

**Figure 3 children-10-01488-f003:**
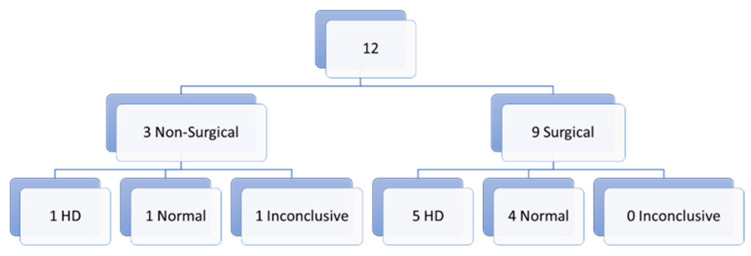
Second biopsies (after a first inconclusive result).

**Figure 4 children-10-01488-f004:**
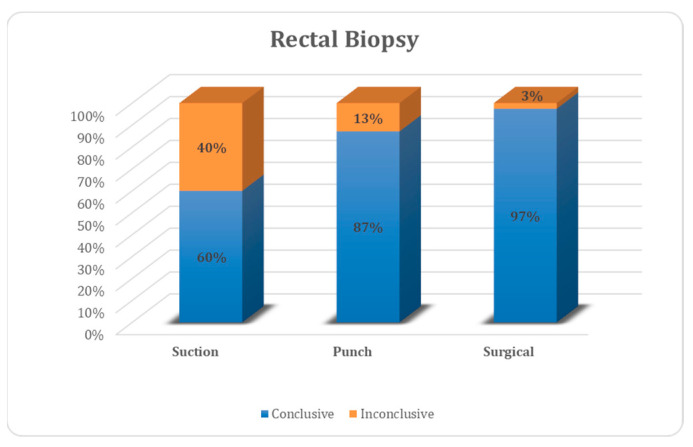
All biopsies.

**Figure 5 children-10-01488-f005:**
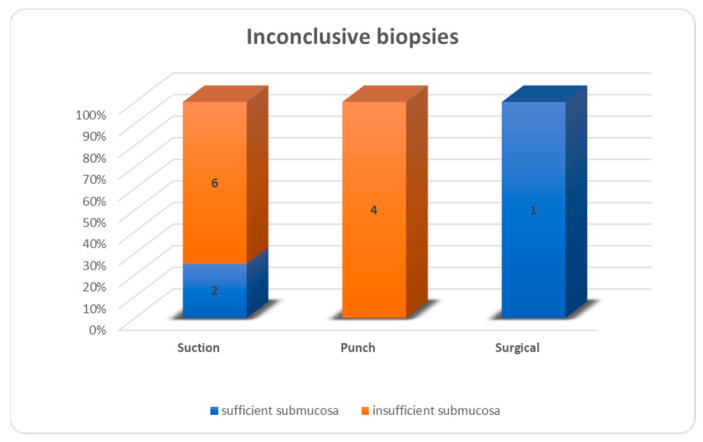
Inconclusive biopsies.

**Figure 6 children-10-01488-f006:**
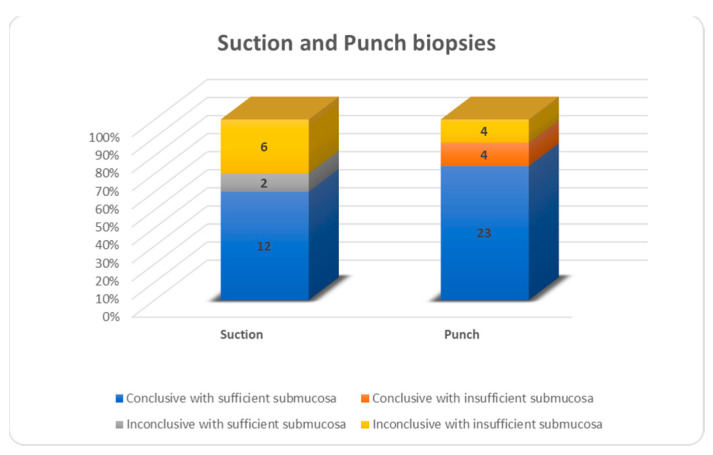
Non-surgical biopsies.

**Table 1 children-10-01488-t001:** Comparison Open versus Non-surgical technique.

Variable	Statistic	Open	Non-Surgical	*p*-Value
CONCLUSIVE				
No	n/N (%)	1/31 (3.23%)	12/51 (23.53%)	0.014
Yes	n/N (%)	30/31 (96.77%)	39/51 (76.47%)	
SUBMUCOSA				
No	n/N (%)	0/31 (0.00%)	14/51 (27.45%)	<0.001
Yes	n/N (%)	31/31 (100.00%)	37/51 (72.55%)	

Variables presented with percentages are analysed using a Fishers Exact test.

**Table 2 children-10-01488-t002:** Comparison of the three techniques.

						Pairwise Comparisons		
Variable	Statistic	Open (1)	Punch (2)	Suction (3)	*p*-Value	1 vs. 2	1 vs. 3	2 vs. 3
CONCLUSIVE								
No	n/N (%)	1/31 (3.23%)	4/31 (12.90%)	8/20 (40.00%)	0.003	0.354	0.001	0.042
Yes	n/N (%)	30/31 (96.77%)	27/31 (87.10%)	12/20 (60.00%)				
SUBMUCOSA								
No	n/N (%)	0/31 (0.00%)	8/31 (25.81%)	6/20 (30.00%)	0.001	0.005	0.002	0.758
Yes	n/N (%)	31/31 (100.00%)	23/31 (74.19%)	14/20 (70.00%)				

Variables presented with percentages are analysed using a Fishers Exact test.

## Data Availability

The data are not publicly available but can be provided on request from the corresponding author.

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
