# Peer review of "Rectal Biopsy for Hirschsprung’s Disease: A Multicentre Study Involving Biopsy Technique, Pathology and Complications"

_children, 2023, doi:10.3390/children10091488_

Round 1
Reviewer 1 Report
The present study highlights a multicentric experience on the technique, pathology and complications of rectal biopsy performed in children suspected to have HD.
In this study, the authors have reviewed the above information from five hospitals. All patients in which rectal biopsies were taken to diagnose Hirschsprung’s disease over two years (2020-2021) were included. 82 biopsies: 20 suction (24,4%), 31 punch (37.8%) and 31 open biopsies (37,8%) were taken. Of all biopsies 69 were conclusive (84,2%), 13 were not (15,8%). In the suction biopsy group 60% were conclusive, 40% not, for punch biopsy 87% and 13% respectively and for open biopsy 97% and 3%. Inconclusive results were due to insufficient submucosa in 6/8 suction biopsies, 4/4 punch and 0/1 open biopsies.
I would like to congratulate the authors for their work. The study has merit and will be of interest to the readers. However, there are a few concerns:
Abstract: well written. No changes needed.
Introduction: What was your hypothesis? Please write your hypothesis in 2-3 lines at the end of the Introduction section.
Methods: No statistical tests were used. No statistical comparison done between the different techniques. The observations of the study could be due to chance. Please introduce statistical comparisons.
-As per the findings of study, only a single biopsy is taken in all techniques. If two biopsies are taken, the sensitivity of non-surgical techniques could improve. Please explain.
Results: I am really skeptical about the results. Different stains have different sensitivity and specificity. Therefore, this could bias the results. Please discuss this.
Discussion: Please expand the limitations section- only single biopsy instead of multiple, variable sensitivity of different stains, etc.
Moderate editing needed
Author Response
Dear,
Thank you for your review report. We took your concerns in considerations. In red, you will find the adjusted part of the manuscript.
Introduction: hypothesis was added
Methods: Statistical analysis was added
The discussion section is adjusted with your concerns in mind.
We hope this will be sufficient for publication.
Sincerely
Gil

Reviewer 2 Report
Hirschsprung’s disease (HD) is a congenital condition of the enteric nervous system. The disordered caudal migration of neural crest cells results in a lack of intrinsic innervation (neuronal ganglion cells and enteric glial cells) in the affected intestine. Diagnosis of the disease can be difficult, in particular in premature children. Histology is still the golden standard to diagnose HD. There are a variety of different rectal biopsy techniques: suction biopsy, punch biopsy and open biopsy.
The title and content of the article represent a topic of real interest worldwide.
The subject of the exposed article is one of real global interest given the heterogeneity in the specialized literature regarding the way to perform the biopsy in the case of patients with Hirshprung's disease. Although the analyzed group is small in scope, I consider the results obtained to be optimal with precise directives regarding the most conclusive way to perform the biopsy, but still, the analysis group being quantitatively reduced, the results cannot be interpreted as a gold standard. In the future, a more extensive analysis, on a quantitatively significant batch of patients, could lead to the implementation of a gold standard biopsy procedure.
The introduction of the article presents originality by proposing a topic with a huge academic potential.
The bibliographic data inserted along the article presents a qualitative chronology. The subject of the article represents a true scientific revolution in its field.
The material and methods section of the article presents a quantitative and qualitative exposition of the research plan, respectively a good reproducibility in order to develop other studies with this theme.
The results of the article present a logical and chronological exposition outlining qualitative aspects of the benefit. The figures and tables keep a specific chronology throughout their exposition, presenting qualitative aspects related to the subject of the article.
The topic of the article is a real interest for the future with major importance in this field. I consider it necessary to develop new studies on this subject and implement them on a population scale. The article presents an important research point with an optimal linguistic exposition, having an exponential potential for the future. This present article is written in a clear and concise manner.
The article presents originality, with an optimal literary exposition, representing a topic of real interest for the future with objective results at the research level. The article represents a launching platform in its field and from the point of view of the characteristics it is included for publication.
Author Response
Dear,
Thank you for your review report.
In red you will find the adjusted part of our manuscript.
We hope this will be sufficient for publication.
Sincerely
Gil

Reviewer 3 Report
well written study. i have suggestions.
1- define aim
2- define more about statistical method?
3-("Correlation of bowel wall thickening seen using computerized tomography with colonoscopies: A preliminary study." Surgical Laparoscopy Endoscopy & Percutaneous Techniques 27.3 (2017): 154-157.) suggested study for the references
Author Response
Dear,
Thank you for your review report. We took in consideration your remarks.
In red you will find the adjusted part of our manuscript.
Introduction: hypothesis was added
Methods: Statistical analysis was added
We hope this will be sufficient for publication.
Sincerely
Gil

Round 2
Reviewer 1 Report
I would like to congratulate the authors for their work. In the revised manuscript, all my comments have been addressed. The overall scientific quality of the manuscript has improved significantly. The study has merit and will be of interest to our readers.
Minor editing is needed